# OpenReview forum: "DIVER: Diving Deeper into Distilled Data via Expressive Semantic Recovery"
_ICML.cc/2026/Conference — ICML 2026 regular_

### Official Review · Reviewer_JhrL · 2026-03-12

**Soundness:** 2
**Presentation:** 3
**Significance:** 2
**Originality:** 2
**Overall Recommendation:** 4
**Confidence:** 3

**Summary:**

This paper proposes DIVER, a two-stage dataset distillation framework designed to tackle the
problem of traditional single-stage methods overfitting to specific architectures, which degrades
cross-architecture generalization. In its first stage, DIVER employs standard distillation techniques
to generate initial distilled data. The second stage, titled Diving into Distilled Data (DDD),
leverages a pre-trained diffusion model to refine this data using three main strategies: semantic
inheritance, semantic guidance, and semantic fusion. Specifically, semantic inheritance maps
images to a latent space to filter out architecture-specific noise. Semantic guidance steers the
reverse generative process to preserve original semantics; and semantic fusion introduces
conditional labels at specific generation steps to prevent semantic ambiguity and sustain the
guidance information.

**Compliance With Llm Reviewing Policy:**

Affirmed.

**Final Justification:**

Thank you for your rebuttal. My questions have been fully addressed. I have increased my score from 3 to 4.

**Key Questions For Authors:**

1. Given that the experiments utilize a DiT model pre-trained on ImageNet-1K, and the
evaluations are also conducted on ImageNet and its subsets, how do the authors rule out the
possibility that the pre-trained model has simply memorized the original data distribution (i.e.,
data leakage)?
2. The core of dataset distillation is to condense information under an extremely tight IPC
(Images Per Class) budget. Why are so many of the classification features originally learned on
the ConvNet lost after reconstruction by a powerful diffusion model (as shown in Table 9)?
3. As said in weakness part, how does DIVER compare in terms of total time and computational
resources against other recent state-of-the-art methods that are entirely based on generative priors
or pure diffusion models?

**Limitations:**

Yes

**Strengths And Weaknesses:**

Strengths
1. The paper points out a critical point in current dataset distillation: traditional single-stage  methods overfit to specific architectures, which restricts their cross-architecture generalization capabilities.
2. This work proposes a novel framework that combines classic dataset distillation methods with pre-trained diffusion models. By utilizing three specific modules (semantic inheritance, semantic guidance, and semantic fusion), it effectively unlocks suppressed high-level semantics.
3. Experimental results across multiple ImageNet subsets demonstrate that this method genuinely improves the generalization performance of traditional approaches on unseen, heterogeneous architectures.

Weaknesses
1. There is a major flaw in the experimental setup. The authors use DiT and VAE models pre-trained on ImageNet-1K, yet the evaluation tasks are conducted precisely on ImageNet and its subsets. This severely weakens the reliability of the experimental conclusions.
2. While the method improves cross-architecture performance, its distillation performance on the original architecture drops significantly. For example, in Table 9, when IPC is 10, the accuracy of the DM method on the ImageFruit dataset falls from 26.2 to 18.4.
3. The paper emphasize in the contributions that the method is highly efficient, saying that it takes “only 2.48 seconds to process a single image.” However, this timeframe solely covers the synthesis time using DiT in the second stage. For a “two-stage” distillation framework, the classic methods used in the first stage (like MTT or DM) typically require extremely high  computational and time costs for optimization.

---

> ### Author Rebuttal · Authors · 2026-03-30
>
> We sincerely appreciate your valuable and insightful suggestions. Our responses are provided below.
>
> **(W1 & Q1) How do the authors rule out the possibility that the pre-trained model has simply memorized the original data distribution (i.e., data leakage)?**
>
> Table 7 demonstrates the robustness of DIVER across different diffusion models. Since SD-v1.5 is trained on the more general LAION dataset rather than the target-domain ImageNet dataset, we still observe a substantial improvement over classical DD (26.5% vs. 20.7%), which validates the generality of our method rather than attributing the gains merely to “data leakage”. Furthermore, by fine-tuning SD-v1.5 on the target-domain dataset to obtain SD-v1.5* and then applying DIVER, we achieve additional performance gains (31.6% vs. 26.5%), indicating that memorizing the target-domain data distribution is indeed beneficial.
>
> | DD       | SD-V1.5  | SD-V1.5* |
> | :------- | :------: | -------- |
> | 20.7±1.8 | 26.5±2.1 | 31.6±1.6 |
>
> **(W2 & Q2) Why are so many of the classification features originally learned on the ConvNet lost after reconstruction by a powerful diffusion model (as shown in Table 9)?**
>
> In real-world scenarios, we often face diverse architectural choices and performance requirements. Therefore, cross-architecture generalization is more important than distillation performance on a specific architecture (ConvNet). Previous studies have shown that the generalization performance of classic DD is largely hampered by high-frequency artifacts introduced by ConvNet [1] and the lack of realism caused by its coupled optimization strategy [2]. As illustrated in Fig. 7 of the Appendix, images directly reconstructed by the VAE are clearer and free of such artifacts, which leads to a drop in ConvNet performance. Furthermore, the semantic restoration provided by diffusion alleviates the blurriness induced by the optimization strategy, causing ConvNet performance to decline even further. The results and analysis in Table 8 and Lines 377–406 quantitatively confirm the reasons for this performance degradation.
>
> Table 8, together with Fig. 7, clearly illustrates this trade-off. After VAE reconstruction, artifacts in the distilled images are partially removed, leading to a corresponding and reasonable decline in ConvNet performance, while CrossArch generalization improves. After the reverse process injects realistic semantic information, the non-interpretable representations induced by conventional DD methods are broken. Meanwhile, our semantic revival module preserves the original generalization information contained in the distilled images. As a result, the final synthtic images exhibit a further decrease in ConvNet performance but a continued improvement in generalization.
>
> | Arc.     | Distilled | Reconstructe (VAE) |   Ours   |
> | -------- | :-------: | :-------------------: | :------: |
> | ConvNet  | 37.1±1.3  |       31.6±0.7        | 28.3±1.1 |
> | CrossArc | 16.1±1.4  |       21.8±1.9        | 26.7±1.7 |
>
> **(W3 & Q3) How does DIVER compare in terms of total time and computational resources against other recent state-of-the-art methods that are entirely based on generative priors or pure diffusion models?**
>
> In Stage 2, our method is comparable to diffusion-based approaches such as MGD$^3$ in both runtime and computational cost, requiring about **2.4s per image and 4 GB of memory overhead**. As you insightfully observed, Stage 1 incurs additional time and resource costs due to the optimization process of classical DD. For example, under the 10 IPC setting on ImageWoof, MTT requires **11 hours and 21 GB of GPU memory**. However, we would like to emphasize that our primary goal is to serve as a plug-in refinement framework built upon existing distilled datasets and to overcome the generalization disadvantages of classic DD. Therefore, Stage 1 is not the main focus of our paper. Certainly, in future work, we aim to directly integrate classical DD strategies into the reverse sampling process of diffusion, which may help reduce the resource consumption in Stage 1.
>
> Thank you again for your constructive review. If you think of any last minute questions, we'll happily do our best to answer before the end of the discussion period.
>
> [1] Generalizing Dataset Distillation via Deep Generative Prior, CVPR 2023.
>
> [2] On the Diversity and Realism of Distilled Dataset: An Efficient Dataset Distillation Paradigm, CVPR 2024.

---

> > ### Author Rebuttal · Reviewer_JhrL · 2026-04-04
> >
> > Thank you for your rebuttal. My questions have been fully addressed. I have increased my score from 3 to 4.

---

> > > ### Author Response · Authors · 2026-04-04
> > >
> > > Thank you for your follow-up and for confirming that the concerns you raised have been properly addressed. We also deeply appreciate the time and attention you have devoted to evaluating our response.

---

### Official Review · Reviewer_qdpE · 2026-03-13

**Soundness:** 2
**Presentation:** 3
**Significance:** 2
**Originality:** 2
**Overall Recommendation:** 2
**Confidence:** 5

**Summary:**

This paper proposes a dual-stage post-processing framework that takes distilled datasets obtained from classical dataset distillation methods as input and performs semantic recovery using a pre-trained VAE and diffusion model. The goal is to improve cross-architecture generalization of the distilled data. Using their proposed algorithm, DIVER, they used DIVER as plug-and-play algorithm to existing works. By applying DIVER, it shown consistent performance improvement across algorithm, dataset.

**Compliance With Llm Reviewing Policy:**

Affirmed.

**Final Justification:**

After considering both the paper and the authors’ rebuttal, I shift toward my final recommendation of weak reject to reject.

My overall assessment remains negative because the main weaknesses were not sufficiently resolved. In terms of originality and significance, the method is ultimately a refinement framework that augments classical dataset distillation with generative priors. Given this design, I believe the most meaningful point of comparison should be diffusion/generative dataset distillation methods, rather than primarily classical DD baselines. On that axis, the paper’s positioning remains unclear to me. The rebuttal argues that DIVER is distinct because it is applied after distilled images are produced, but this did not fully address my concern: at a fundamental level, the method still uses a generative model to improve the distilled data, so the conceptual boundary with existing diffusion-based approaches remains insufficiently justified.

My reading remains that the method largely depends on the quality and information content of the underlying classical distilled set, meaning that its performance ceiling is still inherited from classical DD. If so, the practical significance of adding a generative prior appears limited, especially since the reported gains over stronger generative baselines are not especially substantial. More broadly, if the motivation is to overcome the limitations of classical DD by introducing generative modeling, then I do not find it convincing to simultaneously downplay direct comparison with diffusion-based baselines.
The rebuttal therefore did not change my evaluation. It clarified the authors’ perspective, but it did not resolve my main concerns about mechanism, positioning, and evaluation; if anything, it reinforced my view that the paper’s contribution is narrower than the framing suggests. Overall, while the paper has merit as a practical refinement on top of classical DD, I do not currently find the conceptual novelty, empirical positioning, or practical significance strong enough for acceptance.

**Key Questions For Authors:**

1. In table 2, The performance ranking remains with applying DIVER (For example, DD < DC <MTT persistss DD + DIVER < DC + DIVER < MTT +DIVER). Is it because of the gnerrated quality of the image? Meaning, is it because the fidelity of generated images impact to the performance? Why I question is because DIVER exploits the DD images as regularlizer in diffusion process.

2. For me, this paper is interesting because it allows classical optimization based DD algorithm works in generative area. However, how does DIVER can be applied in large-scale / high-resolution dataset? existing optimization based algorithm cannot handle this data. If so, does DIVER shares this limitation?

3. Comparison to existing works. I think this paper should do extensive comparisoin with existing diffusion-based baseline, not existing DD algorithm. For example, [1] shares similar concept.

If the authors respond to these question. I am willing to raise score.

[1] DIFFUSION MODELS AS DATASET DISTILLATION PRIORS, Su et al.

**Limitations:**

yes

**Strengths And Weaknesses:**

> Soundness:

While the reported improvements over classical dataset distillation baselines are consistent with the paper’s positioning, the comparison with generative/diffusion-based methods is limited. In addition, the mechanism behind the observed gains—particularly why improvements occur on artifact-heavy distilled images—remains insufficiently analyzed.

> Presentation:

The overall structure and description of the method are clear and easy to follow.

> Significance:

Generative priors and diffusion models themselves are not new, and the main contribution lies in a plug-in refinement applied to existing dataset distillation outputs. As a result, the conceptual impact appears somewhat limited.

> Originality:

Modest. Refining existing distilled images with a diffusion prior to improve cross-architecture performance is an interesting direction, but it does not constitute a fundamentally new formulation of dataset distillation.

---

> ### Author Rebuttal · Authors · 2026-03-30
>
> We sincerely appreciate your valuable and insightful suggestions. Our responses are provided below.
>
> **(W1) Analysis of the gain mechanism**
>
> As shown in Fig. 7, the VAE primarily filters out **architecture-specific “noise”** (This stems from its ability to compress high-dimensional images into a low-dimensional space and then reconstruct them into images [1]), whereas the diffusion model **recoveries semantic**, making the images more realistic. Their joint effect enhances generalization performance.
>
> The following table (MTT with IPC 10) illustrates the trade-off between distillation performance and generalization throughout the process of our framework. It also supports our explanation for the improvement on artifact-heavy images: (1) the VAE filters out high-frequency "noise" which has been proven to be detrimental to generalization in GlaD, and (2) diffusion model contributes higher fidelity which has been proven beneficial for generalization in RDED.
>
> | Arc.     | Distilled | Reconstructed | Ours |
> | -------- | --------- | ------------- | ---- |
> | ConvNet  | 37.1      | 31.6          | 28.3 |
> | CrossArc | 16.1      | 21.8          | 26.7 |
>
> **(Q1) The performance ranking remains with applying DIVER**
>
> The performance of the combined DIVER is indeed related to the image quality of the underlying DD. The design of the DIVER cannot ignore the limitations of the classic DD method itself and smooth out their differences.
>
> Specifically, our objective is to **improve the fidelity of synthetic images while preserving the information from the distilled images**, which is also the original motivation behind the design of SI and SG. Therefore, when the images distilled by raw MTT exhibit stronger performance, the combination with DIVER is more likely to retain their intended information, and the resulting images may achieve better generalization. In essence, fidelity is introduced to release the generalization capability that is suppressed by the abstract nature of images generated by classical DD. Consequently, the final performance is jointly determined by the information contained in the images distilled itself by classical DD and by diffusion’s ability to make them more realistic, which align with the views of GLaD and RDED.
>
> **(Q2) Applied in large-scale / high-resolution dataset**
>
> In recent years, diffusion-based methods have shown a strong ability to generate more realistic images with better generalization, which has gradually led traditional DD to be overlooked. The motivation of DIVER is to draw renewed attention to these classical approaches and to narrow the gap between them and diffusion-based methods. We have made every effort to extend it to ImageNet-1k, as reported in Tables 3 and 4, where it achieves promising results. Looking ahead, we hope to directly incorporate the strategies of classical DD into the reverse process, thereby further broadening its applicability.
>
> Meanwhile, classic DD is gradually becoming capable of scaling to large-scale/high-resolution ImageNet-1k, such as TESLA [2] and FADRM [3]. We apply DIVER to them and achieve consistent performance gains.
>
> | IPC  | TESLA | TESLA+Ours | FADRM | FADRM+Ours |
> | ---- | :---: | :--------: | :---: | :--------: |
> | 10   | 17.8  |    22.4    | 50.9  |    53.1    |
> | 50   | 27.9  |    29.3    | 61.2  |    62.5    |
>
>
>
> **(Q3) Comparison to existing works**
>
> Classical DD and diffusion-based DD methods like DAP [4] represent two distinct branches of research. Due to inherent limitations in classical DD, comparing them directly may be considered inequitable. **Recent studies on classical DD (such as NCFM (CVPR 2025), EDF (CVPR 2025), HDD (NeurIPS 2025) [5], etc.) have not included diffusion-based methods as baselines**.
>
> We have also made substantial efforts to integrate our approach into diffusion-based methods. We additionally include DAP [4] and IGD [6] as baselines on ImageNet-1k. The results show that, when combined with MGD$^3$ and Minimax fine-tuning, our method achieves performance comparable to prior state-of-the-art works before 2026. More importantly, we expect DIVER to be more broadly compatible with other diffusion-based methods in the future.
>
> | IPC  | MGD$^3$ | IGD  | DAP  | MGD$^3$+Ours | Minimax+IGD | Minimax+MGD$^3$+Ours |
> | :--- | :----: | :--: | ---- | :---------: | :---------: | :-----------------: |
> | 10   |  45.8  | 45.5 | 49.1 |    46.4     |    46.2     |      **46.9**       |
> | 50   |  60.2  | 59.8 | 62.7 |    60.6     |    60.3     |      **61.0**       |
>
>
> [1] An Introduction to Variational Autoencoders. Foundations and Trends in Machine Learning, 2019.
>
> [2] Scaling Up Dataset Distillation to ImageNet-1K with Constant Memory, ICML 2023.
>
> [3] FADRM: Fast and Accurate Data Residual Matching for Dataset Distillation, NeurIPS 2025.
>
> [4] Diffusion Models as Dataset Distillation Priors, ICLR 2026.
>
> [5] Hyperbolic Dataset Distillation, NeurIPS 2025.
>
> [6] Influence-Guided Diffusion for Dataset Distillation, ICLR 2025.

---

> > ### Author Rebuttal · Reviewer_qdpE · 2026-04-02
> >
> > Thank you for the detailed response. While I appreciate the additional experiments and clarifications, my core concerns remain largely unresolved. The explanation of the gain mechanism is still qualitative and does not sufficiently disentangle the respective roles of fidelity, information preservation, and the contribution of each stage, leaving the observed performance trends under-analyzed. Regarding scalability, although additional ImageNet-1k results are provided, the fundamental question of whether the method inherits the limitations of classical dataset distillation remains unaddressed. Most importantly, the comparison to prior work is still insufficient: the rebuttal does not directly engage with the conceptual similarity to existing diffusion-based approaches, nor does it convincingly justify the choice of baselines. Overall, while the response partially clarifies certain aspects, it does not adequately address the main concerns about mechanism, positioning, and evaluation.
> >
> > ---
> >
> > response to rebuttal
> >
> > Thank you for the additional clarification. I still remain unconvinced on the paper’s positioning for the following reasons.
> >
> > First, the distinction from diffusion-based dataset distillation remains unclear to me. At a fundamental level, DIVER still relies on a generative prior to refine distilled images. From that perspective, it can also be viewed as augmenting classical DD with a diffusion-based generation stage. For this reason, I do not find it persuasive to argue that comparisons to diffusion-based baselines are conceptually inappropriate or unfair.
> >
> > Second, the method still appears to inherit the core limitation of classical DD: its final performance is bounded by the information already contained in the distilled set. If diffusion is introduced precisely to alleviate that limitation, then it is difficult to simultaneously argue that diffusion-based baselines are not the right point of comparison. This makes the current empirical positioning hard to follow.
> >
> > Third, while the plug-in stage itself may be raw-data-free, I do not find this distinction compelling enough to justify the baseline choice in practice, especially when some of the strongest supporting results rely on distilled sets produced by methods such as FADRM, which themselves are optimized from the original data.
> >
> > Overall, although the additional response clarifies the authors’ perspective, my main concerns about positioning and evaluation remain unresolved, and I do not currently see sufficient reason to raise my score.

---

> > > ### Author Response · Authors · 2026-04-03
> > >
> > > Thank you for your constructive feedback. We apologize that our previous response did not fully address your concerns. Below, we provide further clarification.
> > >
> > >
> > >
> > > **1. The explanation of the gain mechanism**
> > >
> > > A key limitation of classical distilled datasets is their insufficient generalization ability, mainly due to **(1) overfitting to architecture-specific high-frequency "noise"**, and **(2) image artifacts and abstraction caused by pixel-level optimization**.
> > >
> > > Our VAE (SI) is used to solve (1), and SG and SF are used to solve (2), as explained below:
> > >
> > > **（1）Why VAE (SI) suppresses high-frequency "noise"**
> > >
> > > VAE performs lossy compression: high-frequency "noise" that are hard to explain and inconsistent with the prior are treated as “non-essential” and get discarded or averaged out.
> > >
> > > The standard VAE optimizes the ELBO:
> > > $$
> > > \max\ \mathbb{E}_{q(z|x)}[\log p(x|z)]\-\\beta\\mathrm{KL}(q(z|x)\|p(z))
> > > $$
> > > The first term encourages accurate reconstruction of $x$ from $z$. The second term pushes the posterior $q(z|x)$ to stay close to the prior $q(z)$  (often a standard Gaussian).
> > >
> > > For high-frequency “noise" of distilled images, encoding them precisely often requires \(q(z|x)\) to become sharp or deviate from Gaussian along some dimensions, which **increases the KL**. But their contribution to “generalizable semantics” is small (and may even be harmful for downstream learning), so under the optimal trade-off the model tends to **not encode them**.
> > >
> > > As a result of **Fig. 7 (right)**, the VAE drops them as “high cost and low benefit” information, the decoded image naturally looks cleaner. As shown in Table 8, images directly reconstructed by the VAE lead to lower ConvNet performance but better generalization, which further supports our conclusion.
> > >
> > > **(2) Diffusion enhances fidelity while preserving information**
> > >
> > > Images reconstructed by the VAE remain abstract and unrealistic. They often deviate from the true manifold (see Fig. 6), thus impairing generalization. Therefore, the role of diffusion is to bring the reconstructed images back to the real data manifold while preserving the valuable information encoded by the VAE in (1).
> > >
> > > For **real data manifold**, this is determined by the characteristics of the diffusion projecting noise to the real data distribution.
> > >
> > > For **preserving the valuable information**, our SG encourages the denoised latent to move toward the VAE-encoded latent in (1), thereby preserving the necessary information $\mathcal{G}_t = (\hat{z}_t - z_0)^2$. SF further prevents overfitting to such information beyond the real data distribution, thereby preserving image fidelity (see Fig. 4 right).
> > >
> > > **2. The comparison to prior work**
> > >
> > > We agree with your concern that our method has not been adequately compared with diffusion-based approaches, mainly for two reasons:
> > >
> > > (1) Diffusion-based methods **still have access to the original dataset, which gives them a higher upper bound**. In contrast, our method is **raw data-free: it only requires the distilled dataset and does not need to complicated match the prototypes (MGD3), influence (IGD), or Mercer similarity (DAP) of the full dataset**. This property is also **beneficial for privacy protection**, since the original dataset is never exposed, while the given distilled dataset is already abstract and unrealistic.
> > >
> > > (2) DIVER as plug-in indeed depends on the quality of images synthesized by existing DD methods, which inherently limits its upper bound. Therefore, we use classic DD as the main baselines, rather than conducting extensive comparisons with diffusion-based methods that require access to the full dataset.
> > >
> > > In summary, the essential difference between DIVER and diffusion-based methods is that diffusion-based methods exploit information from the full dataset, whereas DIVER  relies on information contained in distilled images. If the distilled images are of sufficiently high quality, our method can even outperform diffusion-based approaches.
> > >
> > >
> > > **response to further discuss**
> > >
> > > We sincerely thank you again for your valuable comments. What we want to emphasize is that improving the performance of existing distilled datasets is itself meaningful. This is also the first attempt of this kind in the DD community. Of course, I understand that you hope to extend DIVER further and make it broader, and this may be achieved in our future work.
> > >
> > > **I deeply respect your opinion, and I have done my best to clarify our paper and address your concerns. I do not understand why you showed such strong hostility by lowering the score from 3 to 2, increasing the confidence score to 5, and writing such a long final justification. This is a far cry from the situation before the rebuttal, when you affirmed our paper and expressed your willingness to raise the score.**
> > >
> > > **Although I don't know what happened to you during that time or what I did to offend you, I believe that scientific research isn't everything in life, and I still respect you and sincerely wish you all the best.**

---

### Official Review · Reviewer_WveR · 2026-03-13

**Soundness:** 2
**Presentation:** 2
**Significance:** 3
**Originality:** 3
**Overall Recommendation:** 4
**Confidence:** 4

**Summary:**

Existing dataset distillation (DD) methods are typically single-stage and often suffer from overfitting to prior architectures, resulting in suboptimal performance when transferred to heterogeneous architectures. To address this, this paper proposes DIVER, a dual-stage framework. Beyond the initial distillation stage, DIVER introduces a subsequent "Diving into Distilled Data" stage, which employs a diffusion model to refine the distilled data and recover expressive semantic information that is typically over-suppressed during standard distillation. Extensive experiments demonstrate that DIVER outperforms current state-of-the-art (SOTA) methods. Furthermore, due to its training-free and data-free nature, the proposed method incurs relatively low computational and memory costs, making it highly applicable to a wide range of tasks.

**Compliance With Llm Reviewing Policy:**

Affirmed.

**Key Questions For Authors:**

● Impact of Pre-trained Models: The authors utilize pre-trained models within the "Diving into Distilled Data" (DDD) phase. Could you elaborate on how the choice of these models (e.g., architecture, capacity, or domain) influences the final distilled results? To what extent is the method dependent on the specific pre-trained model?
● Alignment with Identified Drawbacks: The introduction outlines several drawbacks of conventional DD methods. Is there a clear, one-to-one correspondence between these specific challenges and the design "tricks" or components introduced in DIVER? A more explicit mapping would significantly enhance the reader's understanding of the motivation.
● Parameter Sensitivity and Optimization: Several hyper-parameters are introduced in the experimental pipeline. How sensitive is the framework to these parameters, and how difficult is it to tune them for different tasks? Are there any heuristics or recommended strategies to facilitate this optimization process?

**Limitations:**

● Experimental Scope: The current experiments are limited to the 10+ ImageNet subset. Given the "data-free" and "model-free" nature of the proposed framework, it is crucial to evaluate its scalability on more diverse or larger-scale datasets. Expanding the experimental scope would better demonstrate the method's application boundaries and robustness in real-world scenarios.

**Strengths And Weaknesses:**

Strengths:
● The paper is well-written, with a logical flow that is easy to follow.
● The concept of introducing an additional refinement stage for distilled data is novel. By decomposing the process into Semantic Inheritance, Guidance, and Fusion phases, the authors produce visually intuitive distilled data, and the empirical results are compelling.
Weaknesses:
● The organization of the figures could be improved. Specifically, Figure 5, which is critical to understanding the core contribution of the paper, is relegated to the appendix. This forces the reader to jump back and forth, hindering the overall readability.
● Further questions and concerns will be discussed in the following section.

---

> ### Author Rebuttal · Authors · 2026-03-30
>
> We sincerely appreciate your valuable and insightful suggestions. Our responses are provided below.
>
> **(W1) Figure 5, which is critical to understanding the core contribution of the paper, is relegated to the appendix**
>
> Thank you for the suggestion. In the Final revision, we will move Fig. 5 to the main paper (Method section) and update cross-references accordingly to improve readability.
>
> **(Q1) Impact of Pre-trained Models**
>
> We extend the results in Table 7 by evaluating DDD with different pre-trained models, including SD-v1.5 trained on the cross-domain dataset LAION-2B, SD-v1.5* further fine-tuned on the target dataset ImageNet, and DiT (diffusion-based) and SiT (flow-based) trained on target-domain data. We observe that SD-v1.5 still substantially outperforms vanilla DD (26.5% vs. 20.7%), suggesting that DIVER does not strongly depend on models pre-trained on target-domain data. Further fine-tuning on the target domain (SD-v1.5\*) yields a notable additional gain (31.6% vs. 26.5%), indicating that target-domain information is beneficial to our method. SiT performs slightly worse than the diffusion-based DiT (33.1% vs. 34.3%). Finally, compared with the U-Net (SD-v1.5\*), transformer-based architectures (DiT and SiT) appear to be more compatible with our strategy (33.1/34.3% vs. 31.6%).
>
>
>
> |    DD    | SD-V1.5  | SD-V1.5* | DiT      | SiT      |
> | :------: | -------- | -------- | -------- | -------- |
> | 20.7±1.8 | 26.5±2.1 | 31.6±1.6 | 34.3±1.4 | 33.1±2.3 |
>
>
>
> **(Q2) Alignment between identified DD drawbacks and DIVER design components**
>
> Yes, each of our component designs and the mitigation of the shortcomings of classic DD are based on corresponding relationships. SI is designed to filter out architecture-specific “noise” （Refer to the reconstructed image in Fig. 7） that hinders cross-architecture generalization [1]. SG aims to preserve the information in distilled images while releasing authentic visual semantics, whose realism is beneficial to generalization performance [2]. SF, inspired by prior guided diffusion methods [3], is used to reduce artifacts and further enhance the realism of synthesized images. We will also update this explanation in detail in the final manuscript.
>
>
>
> **(Q3) Parameter Sensitivity and Optimization**
>
> As shown in Fig. 3 (left and medium), we primarily use grid search to select hyper-parameters. The optimal parameters for different datasets differ only slightly. To simplify the complex search process, we found that using $\gamma=0.1$ and $t_f=25$ is sufficient. Furthermore, we observed in Fig. 8 that the maximum semantic change steps $t_f$ differ across images.  Specifically, different images exhibit distinct optimal noise-addition steps. For each image, as the number of forward steps increases, the feature variations (e.g., texture details) gradually intensify. However, this change eventually diminishes and stabilizes, as the initial latent variables asymptotically converge to the same Gaussian distribution. These can serve as a heuristic for recommendation strategies.
>
>
>
> **(L1) Experimental Scope**
>
> Since DIVER is a plug-in framework, its applicability is largely constrained by the operational boundaries of existing DD methods. The main experiments were conducted on an ImageNet subset primarily because classical DD approaches, such as MTT, rely on optimization in pixel space and thus incur substantial GPU memory and computational costs on large-scale datasets. Nevertheless, we have also extended DIVER to ImageNet-1k, for example through SRe$^2$L (Table 3) and MGD$^3$ (Table 4). More importantly, our goal is for DIVER to serve as a bridge between classical DD methods and diffusion-based approaches, with the potential to scale to broader applications in the future.
>
>
>
> [1] Generalizing Dataset Distillation via Deep Generative Prior, CVPR 2023.
>
> [2] On the Diversity and Realism of Distilled Dataset: An Efficient Dataset Distillation Paradigm, CVPR 2024.
>
> [3] Universal Guidance for Diffusion Models, CVPR 2023.
>
> Thank you again for your constructive review. If you think of any last minute questions, we'll happily do our best to answer before the end of the discussion period.

---

> > ### Author Rebuttal · Reviewer_WveR · 2026-04-01
> >
> > The response to the questions still need  more work to do. For example the impact of pre-trained models need more examples. Another example is that, more detailed explanations are needed for Q2, etc.

---

> > > ### Author Response · Authors · 2026-04-03
> > >
> > > Thank you for your positive feedback. We apologize that our response did not fully address your concerns. Below, we provide further response.
> > >
> > > **1. Impact of pre-trained DiT model scaling on performance**
> > >
> > > We followed the settings [1] and compared DiT models of different capacities to study the effect of model size on distillation performance, with the patch size fixed at 2.
> > >
> > > We found that larger pre-trained models produce better image quality. As a result, the performance improves as the capacity of the model increases, which also demonstrates the scalability of our method.
> > >
> > > | Model    | Layers | Hidden size | Heads | Acc.         |
> > > | -------- | ------ | ----------- | ----- | ------------ |
> > > | DiT-S/2  | 12     | 384         | 6     | 30.8±1.7     |
> > > | DiT-L/2  | 24     | 1024        | 16    | 33.1±2.0     |
> > > | DiT-XL/2 | 28     | 1152        | 16    | **34.3±1.4** |
> > >
> > > We agree that DIVER still requires more extensions on pre-trained models. In future work, we also plan to explore combining it with recently popular generative models based on representation encoders which discards VAE [2-3].
> > >
> > > **2. Further clarification regarding Q2 (Alignment between identified DD drawbacks and DIVER design components)**
> > >
> > > A key limitation of classical distilled datasets is their insufficient generalization ability, mainly due to **(1) overfitting to architecture-specific high-frequency "noise"**, and (2) **image artifacts and abstraction caused by pixel-level optimization**.
> > >
> > > Our VAE (SI) is used to solve (1), and SG and SF are used to solve (2), as explained below:
> > >
> > > **（1）VAE (SI) filters out "noise"**
> > >
> > > VAE performs lossy compression: high-frequency "noise" that are hard to explain and inconsistent with the prior are treated as “non-essential” and get discarded or averaged out.
> > >
> > > The standard VAE optimizes the ELBO:
> > > $$
> > > \max\; \mathbb{E}_{q(z|x)}[\log p(x|z)]\;-\;\beta\,\mathrm{KL}(q(z|x)\|p(z))
> > > $$
> > > The first term encourages accurate reconstruction of \(x\) from \(z\). The second term pushes the posterior \(q(z|x)\) to stay close to the prior \(p(z)\) (often a standard Gaussian).
> > >
> > > For high-frequency “noise" of distilled images, encoding them precisely often requires \(q(z|x)\) to become sharp or deviate from Gaussian along some dimensions, which **increases the KL**. But their contribution to “generalizable semantics” is small (and may even be harmful for downstream learning), so under the optimal trade-off the model tends to **not encode them**.
> > >
> > > As a result of **Fig. 7 (right)**, the VAE drops them as “high cost and low benefit” information, the decoded image naturally looks cleaner. As shown in Table 8, images directly reconstructed by the VAE lead to lower ConvNet performance but better generalization, which further supports our conclusion.
> > >
> > > **(2) SG and SF enhances fidelity while preserving information**
> > >
> > > Images reconstructed by the VAE remain abstract and unrealistic. They often deviate from the true manifold (see Fig. 6), thus impairing generalization. Therefore, the role of diffusion is to bring the reconstructed images back to the real data manifold while preserving the valuable information encoded by the VAE in (1).
> > >
> > > For **real data manifold**, this is determined by the characteristics of the diffusion-projected noise to the real data distribution.
> > >
> > > For **preserving the valuable information**, our SG encourages the denoised latent to move toward the VAE-encoded latent in (1), thereby preserving the necessary information $\mathcal{G}_t = (\hat{z}_t - z_0)^2$. SF further prevents overfitting to such information beyond the real data distribution, thereby preserving image fidelity (see Fig. 4 right).
> > >
> > > In this way, our entire pipeline removes the architecture-specific noise and abstract representations in classical distilled images while preserving valuable semantic information.
> > >
> > > We sincerely thank you again for your valuable comments and suggestions, which are very helpful for improving the quality of our paper. According to the official policy, only one round of interaction is allowed. Therefore, we are not sure whether we will have the opportunity to revise the acknowledgment or continue the discussion through additional comments. If such further discussion is allowed, we will do our best to address any additional questions before the discussion ends.
> > >
> > > [1] Scalable Diffusion Models with Transformers, ICCV 2023.
> > >
> > > [2] SVG: Latent Diffusion Model without Variational Autoencoder, ICLR 2026.
> > >
> > > [3] Diffusion Transformers with Representation Autoencoders, ICLR 2026.

---

### Official Review · Reviewer_SaU9 · 2026-03-17

**Soundness:** 3
**Presentation:** 3
**Significance:** 3
**Originality:** 3
**Overall Recommendation:** 5
**Confidence:** 4

**Summary:**

This paper proposes a dual-stage distillation framework termed DIVER, which can easily improve cross-architecture generalization performance in a raw data-free and training-free manner. DIVER is simple and effective, and it has the potential to have a profound impact on the DD community as a new paradigm.

**Compliance With Llm Reviewing Policy:**

Affirmed.

**Final Justification:**

This paper proposes a simple and efficient dual-stage distillation framework termed DIVER that does not rely on raw data or training, and has the potential to serve as a new paradigm with a profound impact on the data distillation community. The authors' responses have addressed my concerns, and I therefore decide to increase the score to 5.

**Key Questions For Authors:**

1. Could the author explain the fundamental differences between SF and SG in order to better understand their roles in the reverse process?
2. Can the author provide several pieces of evidence regarding VAE filtering “noise”?
3. While the images synthesized by DIVER is quite interesting, seemingly inheriting the texture of distilled data, the absence of SF doesn't explain why artifacts or image distortion would occur. Could you provide a more detailed analysis of  w/o SF?

**Limitations:**

Yes

**Strengths And Weaknesses:**

Strengths:
1. Simple and Effective. The method is simple and has achieved consistent incremental results.
2. Efficient and Practical. DIVER can be used as a plugin to improve the performance of existing distilled datasets with minimal overhead (comparable to raw DiT), making it easy and efficient to apply.
3. Privacy Protection. This paradigm avoids approaching the original dataset, which is crucial for privacy protection.
4. Sufficient experiments. The paper discusses results across multiple datasets, methods, and diffusion models. The experiments consistently outperform the classic DD.

Weakness:
1. SI is easy to understand, but SF and SG are both implemented in the SP phase (Figure 2), so to me they don't seem to be very different.
2. The authors emphasize that VAE filters “noise”, but this has not been verified.
3. The cause of artifacts in w/o SF (Figure 4) is not clearly explained.
4. In Figure 3 (medium), 19.88 is reported with two decimal places, while the other numbers are reported with only one decimal place.

---

> ### Author Rebuttal · Authors · 2026-03-30
>
> We sincerely appreciate your valuable and insightful suggestions. Our responses are provided below.
>
> **(W1 & Q1) Could the author explain the fundamental differences between SF and SG in order to better understand their roles in the reverse process?**
>
> SG is essentially a function aimed at ensuring that the images generated by diffusion are consistent with the feature representations of the distilled images. SF, on the other hand, is a time stage that enables SG to work, namely SP. In this stage, SF integrates the latent inherited from the distilled images, text semantics (labels), and SG. Fig. 5 in the appendix and the pseudocode in lines 631-652 provide further details to help you better understand the distinction between them.
>
>
>
> **(W2 & Q2) Can the author provide several pieces of evidence regarding VAE filtering “noise”?**
>
> From the results in Figure 7 of the appendix, we notice that the images reconstructed by VAE (direct encoding and decoding without DiT) are "clearer" than those in the distilled dataset. This qualitatively confirms that encoding high-dimensional images into low-dimensional images by VAE does indeed filter out "noise." On the other hand, the experimental results in Table 9 show that the distillation performance of the reconstructed images on ConvNet decreases, which quantitatively confirms that "noise" is filtered (MTT with IPC 10).
>
> | Arc.     | Distilled | Reconstructed | Ours |
> | -------- | --------- | ------------- | ---- |
> | ConvNet  | 37.1      | 31.6          | 28.3 |
> | CrossArc | 16.1      | 21.8          | 26.7 |
>
>
>
> **(W3 & Q3) While the images synthesized by DIVER is quite interesting, seemingly inheriting the texture of distilled data, the absence of SF doesn't explain why artifacts or image distortion would occur. Could you provide a more detailed analysis of w/o SF?**
>
> SF makes the category information clearer and has better fidelity, while its absence leads to blurring and artifacts in synthetic images, which also accounts for the performance gap described in Tab. 6. This phenomenon primarily stems from two factors:
>
> (1) The latent code of the distilled dataset still retains residual architecture-specific patterns, where overfitting exacerbates image distortion and artifacts, inevitably degrading performance.
>
> (2) Full-phase guidance obstructs the injection of conditional category information and also runs the risk of overfitting, ultimately compromising visual fidelity.
>
>
>
> **(W4) In Figure 3 (medium), 19.88 is reported with two decimal places, while the other numbers are reported with only one decimal place.**
>
> We appreciate the reviewer's very detailed observation. And we will revise it in the final version to retain one decimal place.
>
> Thank you again for your constructive review. If you think of any last minute questions, we'll happily do our best to answer before the end of the discussion period.

---

> > ### Author Rebuttal · Reviewer_SaU9 · 2026-04-01
> >
> > My concerns have been addressed, and I have no further comments.

---

> > > ### Author Response · Authors · 2026-04-03
> > >
> > > We sincerely thank you for your further feedback and for confirming that the concerns you raised have been properly addressed in our rebuttal. We also deeply appreciate the time and attention you have devoted to evaluating our response.

---

### Official Review · Reviewer_yhWW · 2026-03-18

**Soundness:** 4
**Presentation:** 3
**Significance:** 3
**Originality:** 3
**Overall Recommendation:** 5
**Confidence:** 5

**Summary:**

This paper utilizes diffusion model to perform a dual-stage refinement of distilled images synthesized from classic DD, thereby improving generalization ability. This innovative paradigm can work without the original dataset and requires very little additional overhead. This has practical implications for privacy protection and efficient learning.

**Compliance With Llm Reviewing Policy:**

Affirmed.

**Key Questions For Authors:**

1. The results in Table 8 show the performance trade-off between CrossArc and ConvNet. Has the author provided a detailed explanation of this?
2. The stable improvements brought by DIVER are obvious, but its underlying working mechanism seems unclear. Can the authors provide further explanation as to why this paradigm works effectively?
3. I'm not sure if my understanding of random* in Table 6 is correct. Does it simply replace the distilled image of the classic DD synthesis with a randomly selected real image?
In conclusion, I think DIVER is simple yet very fun. If the author can address my concerns, I will consider raising my score.

**Limitations:**

Yes.

**Strengths And Weaknesses:**

Strengths
1. DIVER is an efficient paradigm. Based on existing distilled images, it achieves stable generalization improvements with minimal overhead.
2. The paper is well-written and this plugin implementation pattern is easy to follow.
3. The visualizations produced by the synthesized images are interesting and insightful, fundamentally different from traditional diffusion-based methods. This may lay the foundation for a revival of classic DD methods.
4. Extensive main experiments and ablation experiments comparing the method with classic baselines validated its effectiveness.

Weakness
1. While DIVER performs well across architectures, it exhibits a significant performance drop on specific ConvNets（Table 8 and Fig 3 right）. The reasons and implications of this should be explained.
2. Using a diffusion prior to improve generalization performance seems intuitively reasonable。 However, a lack of sound theoretical or experimental evidence may hinder further exploration of the method.

Questions
1. The results in Table 8 show the performance trade-off between CrossArc and ConvNet. Has the author provided a detailed explanation of this?
2. The stable improvements brought by DIVER are obvious, but its underlying working mechanism seems unclear. Can the authors provide further explanation as to why this paradigm works effectively?
3. I'm not sure if my understanding of random* in Table 6 is correct. Does it simply replace the distilled image of the classic DD synthesis with a randomly selected real image?
In conclusion, I think DIVER is simple yet very fun. If the author can address my concerns, I will consider raising my score.

---

> ### Author Rebuttal · Authors · 2026-03-30
>
> Thank you very much for your constructive feedback, the valuable question and your interest in DIVER. Our responses are provided below.
>
> **(W1 & Q1) A significant performance drop on specific ConvNets**
>
> In Fig. 7, the VAE serves to eliminate **architecture-specific “noise”**, effectively filtering out irrelevant details. In contrast, the diffusion model restores semantic information, enhancing the realism of the images. Together, these two components improve generalization performance.
>
> The table below (MTT with IPC 10) illustrates the trade-off between distillation performance and generalization within our framework. It also supports our interpretation of the improvements observed in artifact-heavy images: (1) the VAE suppresses high-frequency “noise,” which prior work (GlaD) has shown to be harmful to generalization; and (2) the diffusion model enhances fidelity, which prior work (RDED) has shown to benefit generalization.
>
> | Arc.     | Distilled | Reconstructed | Ours |
> | -------- | --------- | ------------- | ---- |
> | ConvNet  | 37.1      | 31.6          | 28.3 |
> | CrossArc | 16.1      | 21.8          | 26.7 |
>
> **(W2 & Q2) A lack of sound theoretical or experimental evidence may hinder further exploration of the method**
>
> We primarily leverages the inherent theoretical properties of VAE and guided diffusion to explain our overall pipeline.
>
> (1) The VAE compresses high-dimensional images into a lower-dimensional latent space, inherently preserving semantic content. Latent code is essentially a compressed structured and semantic representation.
>
> (2) Our guided diffusion process (energy-based models, EBM) [1] is adaptively steered by learned feedback mechanisms (specifically, Semantic Guidance related to z) to generate target-specific outputs according to user requirements.
>
> Inspired by GLaD and RDED, which advocate removing high-frequency "noise" while preserving image fidelity to improve generalization, we propose SI and SG to retain the valuable information in the raw distilled images and subsequently use diffusion to enhance their realism. The ablation results in Table 6 further validate the effectiveness of our method.
>
>
>
> **(Q3) The understanding of random**
>
> Yes, "Random*" denotes directly feeding randomly selected images into our pipeline. This experiment is designed to verify that the gain does not come from diffusion alone, but more importantly from the valuable information contained in the distilled images.
>
>
> [1] Universal Guidance for Diffusion Models, CVPR 2023.
>
> Thank you again for your constructive review. If you think of any last minute questions, we'll happily do our best to answer before the end of the discussion period.

---

> > ### Author Rebuttal · Reviewer_yhWW · 2026-04-01
> >
> > The author uses the visualization in Figure 7 to verify that the reduction in "noise" and the performance degradation of ConvNet are intuitive. The explanation of the performance improvement based on the characteristics of VAE and guided diffusion models is also intuitively reasonable. In conclusion, I believe DIVER is a novel DD paradigm that may be impressive, therefore I decided to improve my score.

---

> > > ### Author Response · Authors · 2026-04-03
> > >
> > > We sincerely thank the reviewers for your follow-up evaluation and recognition of our work, and are pleased to see that our rebuttal resolved the relevant issues in a satisfactory manner.

---

### Decision · Program_Chairs · 2026-04-30

**Decision:**

Accept (regular)

**Comment:**

This paper proposes a novel raw data-free and training-free framework called DIVER. This paradigm, as a plugin, can consistently improve the performance of existing distilled datasets and may be further developed by the DD community in the future.

All reviewers unanimously recognized the paper’s innovation, efficiency, practicality, and comprehensive experiments. After the rebuttal, most reviewers (two gave “Accept” and two gave “Weak Accept”) recommended acceptance and acknowledged the authors’ clarifications. Reviewer qdpE’s main concern was whether the method could be more broadly extended to diffusion-based approaches. However, I believe the paper has already conducted extensive experiments to validate the effectiveness of the framework and addressed reviewers’ concerns, and this extra concern seems to be beyond the scope of the current work. Therefore, following the consensus of the majority of reviewers, I recommend acceptance.